# AutoShot: A Short Video Dataset and State-of-the-Art Shot Boundary Detection

## Abstract

The short-form videos have explosive popularity and have dominated the new social media trends. Prevailing short-video platforms, *e.g.*, TikTok, Instagram Reels, and YouTube Shorts, have changed the way we consume and create content. For video content creation and understanding, the shot boundary detection (SBD) is one of the most essential components in various scenarios. In this work, we release a new public Short video sHot bOundary deTection dataset, named SHOT, consisting of 853 complete short videos and 11,606 shot annotations, with 2,716 high quality shot boundary annotations in 200 test videos. Leveraging this new data wealth, we propose to optimize the model design for video SBD, by conducting neural architecture search in a search space encapsulating various advanced 3D ConvNets and Transformers. Our proposed approach, named AutoShot, achieves higher F1 scores than previous state-of-the-art approaches, *e.g.*, outperforming TransNetV2 by 4.2%, when being derived and evaluated on our newly constructed SHOT dataset. Moreover, to validate the generalizability of the AutoShot architecture, we directly evaluate it on another three public datasets: ClipShots, BBC and RAI, and the F1 scores of AutoShot outperform previous state-of-the-art approaches by 1.1%, 0.9% and 1.2%, respectively. The SHOT dataset and code will be released.

## 1 Introduction

Short-form videos have been widely digested among the entire age groups all over the world. The percentage of short videos and video-form ads has an explosive growth in the era of 5G, due to the richer contents, better delivery and more persuasive effects of short videos than the image and text modalities (Wang et al., 2021). This strong trend leads to a significant and urgent demand for a temporally accurate and comprehensive video analysis in addition to a simple video classification category. Shot boundary detection is a fundamental component for temporally comprehensive video analysis and can be a basic block for various tasks, *e.g.*, scene boundary detection (Rao et al., 2020; Chen et al., 2021), video structuring (Wang et al., 2021), and event segmentation (Shou et al., 2021). For instance, rewarded videos can be automated created of desired lengths for different platforms, leveraging the accurate shot boundary detection in the intelligent video creation.

To accelerate the development of video temporal boundary detection, several datasets have been collected with laboriously manual annotation. Conventional shot boundary detection datasets, *e.g.*, BBC Planet Earth Documentary series (Baraldi et al., 2015a) and RAI (Baraldi et al., 2015b), only consist of documentaries or talk shows where the scenes are relatively static. Tang et al. (2018) further contribute a large-scale video shot database, ClipShots, consisting of different types of videos collected from YouTube and Weibo covering more than 20 categories, including sports, TV shows, animals, *etc*. Shou et al. (2021) construct a generic event boundary detection (GEBD) dataset, Kinetics-GEBD, which defines a clip as the moment where humans naturally perceive an event. Since the video lengths of short and conventional videos differ extensively, *i.e.*, 90% short videos of length less than one minute *versus* videos in other datasets having length of 2-60 minutes as shown in Table 1 and Fig. 1 Right, it dramatically leads to significant content, display, temporal dynamics and shot transition differences as shown in Fig. 1 Left. A short video dataset is necessary to accelerate the development and proper evaluation of short video based shot boundary detection.

On the other hand, several endeavors have been made to improve the accuracy of video shot boundary detection (SBD). DeepSBD (Hassanien et al., 2017) firstly applies a deep spatio-temporal ConvNet

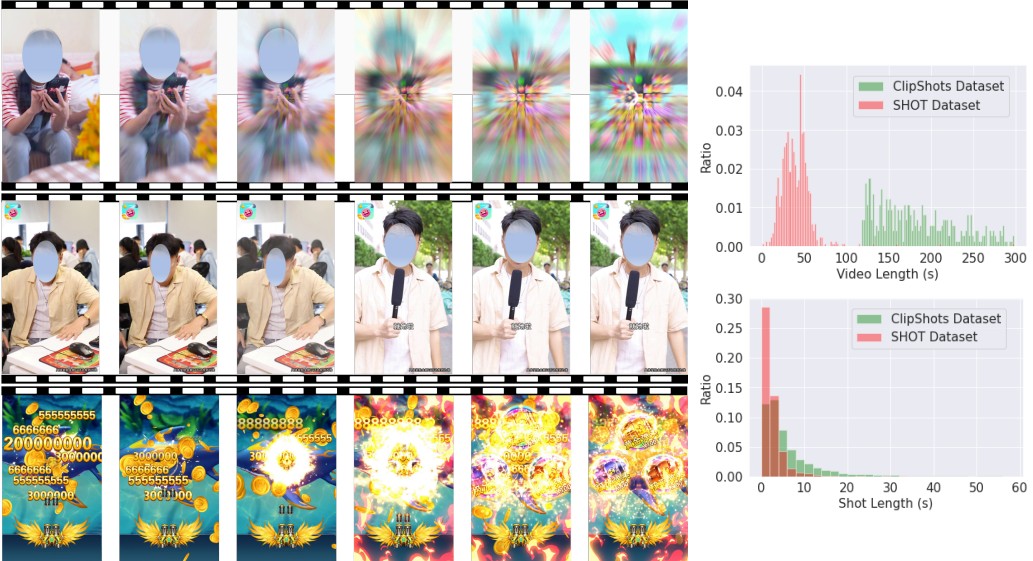

Figure 1: Left: Detecting a shot boundary can be a challenging task in short videos. The shot transition can be a combination of several complicated gradual transitions (the first row) and a quick transition of the subject in two shots (the second row). The visual effect of the intra-shot can vary greatly in game videos (the third row). Right: Video and shot length (s) comparison of test sets in ClipShots and our collected SHOT. There rarely has a video length range overlap between short videos in SHOT and test videos in ClipShots (up). The shot lengths of short videos are within six seconds, while the shot lengths of ClipShots can range from two seconds to 30 seconds (bottom).

to the video SBD. Deep structured model (DSM) (Tang et al., 2018) designs a cascade framework to accelerate the speed of SBD. TransNet (Lokoč et al., 2019) uses dilated convolutional cells (Yu & Koltun, 2016) to process a sequence of resized frames. TransNetV2 (Souček & Lokoč, 2020) incorporates techniques, *e.g.*, convolution kernel factorization (Xie et al., 2018), batch normalization (Ioffe & Szegedy, 2015), skip connection (He et al., 2016), and further improves F1 scores on ClipShots (Tang et al., 2018) and BBC (Baraldi et al., 2015a).

In this work, we firstly collect a short video dataset, named SHOT, consisting of 853 short videos with 11,606 manually shot boundary fine annotations. The 200 test videos with 2,716 shot boundary annotations are labeled by experts with two rounds. Leveraging this new data wealth, we aim to improve the accuracy of video shot boundary detection, by conducting neural architecture search in a search space encapsulating various advanced 3D ConvNets (Qiu et al., 2017) and Transformers (Vaswani et al., 2017). Single path one-shot SuperNet strategy (Guo et al., 2020) and Bayesian optimization (Shahriari et al., 2015) are employed. The searched model, named AutoShot, outperforms TransNetV2 by 4.2% on our SHOT in terms of F1 score, and by 3.5% in terms of precision metric with a fixed recall rate as TransNetV2, respectively. We further evaluate the searched AutoShot architecture on ClipShots, BBC and RAI, and F1 score of AutoShot surpasses previous state-of-the-art approaches by 1.1%, 0.9% and 1.2%, respectively. Our contributions are summarized as follows:

- We collect a short video shot boundary detection dataset (SHOT), which consists of 853 short videos and 11,606 shot boundary annotations. The SHOT will be released and can be employed to advance the development of various short video understanding tasks.

- We design a video shot boundary detection search space encapsulating various advanced 3D ConvNets and Transformers, and build a neural architecture search pipeline for shot boundary detection.

- The searched model, named AutoShot, proves to be a highly competitive shot boundary detection architecture, that significantly outperforms previous state-of-the-art approaches not only on its derived SHOT dataset, but also on other public benchmarks.

To the best of our knowledge, the collected SHOT dataset is the first dataset for short video shot boundary detection, and AutoShot is the firstly specially designed neural architecture search method for shot boundary detection.

## 2 RELATED WORK

Extensive efforts have been made to collect video *shot* boundary detection datasets (Tang et al., 2018; Baraldi et al., 2015a;b; Chakraborty et al., 2022; Jiang et al., 2021; Rashmi & Nagendraswamy, 2021), which have significantly accelerated the development of advanced *shot* boundary detection methods. A specific type of video boundary detection attempts to parse the video into pieces of human actions, where instructional videos with human performing diverse actions are commonly used (Kuehne et al., 2014; Stein & McKenna, 2013; Tang et al., 2019). Another widely studied type of video boundary detection is *scene* boundary detection, where videos are split into several semantically independent clips. Movies and TV episodes are commonly used for *scene* boundary detection, and MovieScenes (Rao et al., 2020) and AdCuepoints (Chen et al., 2021) are two large-scale movie datasets for *scene* boundary detection. Furthermore, Shou et al. (2021) collect a new benchmark, Kinectics-GEBD, for generic *event* boundary detection. For video ads *scene* boundary detection, Wang et al. (2021) recently collect a multi-modal video ads understanding dataset, which involves *scene* boundary detection and multi-modal scene classification. However, *scene* boundary detection is typically based on the pre-extracted *shots* and the evaluation of scene segmentation is *shot*-level instead of frame-level.

The TV series and talk shows have also been used for *shot* boundary detection, where BBC Planet Earth Documentary series (Baraldi et al., 2015a) and RAI (Baraldi et al., 2015b) are two commonly used datasets consists of tens of videos having length from half an hour to one hour. ClipShots (Tang et al., 2018) enhances the conventional *shot* boundary detection datasets by collecting diverse videos from various media platforms, *e.g.*, YouTube and Weibo, and it is one of the most challenging large-scale *shot* boundary detection datasets. The shot transitions in short videos, are quite different from that of movies as shown in Fig. 1 Right and Table 1, and it is extremely necessary that a short video dataset is collected to advance the development of short video *shot* boundary detection. The only short video dataset publicly available so far, to our best knowledge, is the SVD dataset (Jiang et al., 2019); yet that is developed for a completely different task for near-duplicate video retrieval.

The accuracy of *shot* boundary detection has been improved, leveraging the aforementioned high quality datasets and deep learning. Before the deep learning era, PySceneDetect (Castellano, 2022) is a popular *shot* boundary detection library, which relies on conventional features, *e.g.*, changes between frames in the HSV color space. Recent progresses on video boundary detection can be divided into two categories, *scene* boundary detection and *shot* boundary detection. Rao et al. (2020) propose a local-to-global *scene* segmentation framework integrating multi-modal information across clip, segment, and movie. Chen et al. (2021) further propose a shot contrastive self-supervised learning to learn a shot representation that maximizes the similarity between nearby shots compared to randomly selected shots, then apply the learned shot representation for *scene* boundary detection.

The *scene* boundary detection highly depends on the accurate *shot* boundary detection. DeepSBD (Hassanien et al., 2017) predicts a likelihood of transitions in a clip of 16 frames by the C3D network (Tran et al., 2015). DSM (Tang et al., 2018) utilizes a cascade framework to accelerate the speed of *shot* boundary detection. Gygli (Gygli, 2018) constructs a fast *shot* boundary detection without any post-processing. TransNet (Lokoč et al., 2019) uses dilated convolution blocks (Yu & Koltun, 2016) and achieves comparable accuracy as DeepSBD without post-processing. TransNetV2 (Souček & Lokoč, 2020) surpasses previous state-of-the-art approaches with advanced components, *e.g.*, skip connection (He et al., 2016), batch normalization (Ioffe & Szegedy, 2015), kernel factorization (Xie et al., 2018), frame similarities as features and multiple classification heads. Leveraging the progress of 3D ConvNets (Qiu et al., 2017), Transformers (Vaswani et al., 2017) and neural architecture search (Guo et al., 2020), AutoShot automatically identifies the optimal *shot* boundary detection architecture from the designed search space, which achieves better accuracy than previous methods on the collected SHOT, ClipShots, BBC and RAI datasets.

## 3 SHOT: SHORT VIDEO SHOT BOUNDARY DETECTION DATASET

Short video is one of the most prevailing medias in these days because of its richer contents and more vivid effects than its conventional counterparts, *i.e.*, pure text and static picture medias. The easy and affordable access to fast mobile networks in the 5G era accelerates the widespread adoption of short video platforms, *e.g.*, Instagram Reels, YouTube Shorts, and TikTok. With the large number of users and video ads in these main-stream short video platforms, it is critical to advance the current development of video temporal segmentation, especially short video shot boundary detection task, which is a fundamental task for many following semantic understanding tasks.

### 3.1 CHALLENGES OF SHORT VIDEO SHOT BOUNDARY DETECTION

A short video is typically defined as a video of length less than two minutes. The short video length leads to a much easier spread of these videos and more popular short videos than conventional movies of hours long. On the other hand, the short video length forces the whole events to occur in a short time period, which causes much faster pace of events. This in turn leads to a much shorter length of a shot in the short video as shown in Fig. 1 Right, which aggregates the difficulty of short video shot boundary detection.

The giant difference of video lengths between the test sets in the collected SHOT dataset and ClipShots (Tang et al., 2018) is visualized in Fig. 1 Right. Almost all the test short videos have lengths less than 100 seconds, while almost all the test videos in the ClipShots have lengths greater than 120 seconds. The short total video length directly leads to rapid shot transitions in the SHOT dataset as indicated in Fig. 1 Right. Most shot lengths are within five seconds in the test set of SHOT dataset, and the shot lengths of test set in the ClipShots can range from two seconds to 30 seconds. The conventional shot boundary detection dataset may be inappropriate for the development of short video shot boundary detection because of the great difference of video and shot length distributions.

Short video shot boundary detection is much more challenging and difficult as illustrated in Fig. 1 Left and Fig. 2 because the scene of the short video is much more complicated than conventional videos. For instance, the shot transition commonly utilizes a combination of several complicated shot gradual transitions for the persuasive effect in the short video (the first row). The second common challenge is for vertically ternary structured videos (the second row of Fig. 2), where only the middle part of the video changes. The uppermost and lowermost regions display the download link or brand in the video ads. The vertically ternary structured video increases shot boundary detection difficulty greatly due to the relatively small region change in the squeezed content region. The exaggerated expression in the virtual scene causes false alarms in the game video shot boundary detection (the third row). Actually, the game video takes a large proportion of video ads and short videos. Therefore, collecting a short video dataset is nontrivial for the challenging short video shot boundary detection.

### 3.2 SHOT DATASET

To accelerate the study of short video related shot boundary detection, we collect 853 short videos from one of the most widely used short video platforms. The dataset property comparison is listed in Table 1. The total number of frames is 960,794, close to one million frames. The frame-wise shot boundary annotation is a heavy task. The data annotation and quality control strategy are in appendix.

The cases from the SHOT dataset can be found in Fig. 1 Left, Fig. 2, and Fig. 4. The annotation records the start and end frame numbers of each shot, which is visualized by the pink and light/white colors in Fig. 4. All the training and testing short videos, shot boundary annotations, the evaluation metric scripts and the explicit video-level data split will be publicly available.

## 4 AUTOMATED SHOT BOUNDARY DETECTION

### 4.1 SHOT BOUNDARY DETECTION SEARCH SPACE DESIGN

We design a SuperNet based on one of the previous state-of-the-art approaches, TransNetV2 (Souček & Lokoč, 2020), where the feature representational learning network can be considered as a sequence of six factorized dilated deep 3D convolutional neural network (DDCNNV2) blocks. Leveraging the

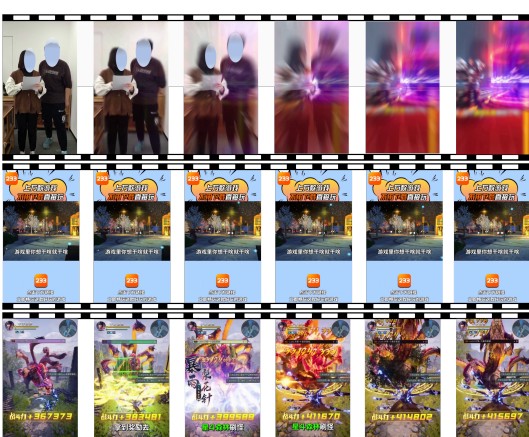

| Dataset | BBC | RAI | Clip. | SHOT |
|---|---|---|---|---|
| Complex Grad. Trans. | ✗ | ✗ | ✗ | ✓ |
| Virt. Scene | ✗ | ✗ | ✗ | ✓ |
| Tern. Video | ✗ | ✗ | ✗ | ✓ |
| Avg. Video Len. (s) | 2945 | 591 | 237 | 39.5 |
| Avg. Shot Len. (s) | 6.57 | 5.65 | 15.34 | 2.59 |

Table 1: Comparison of different short boundary detection datasets, *i.e.*, BBC, RAI, ClipShots and SHOT, *w.r.t.* complex gradual transition, virtual scene, ternary video, average video length and average shot length.

Figure 2: Unique challenges of short video shot boundary detection in SHOT, *e.g.*, a combination of complicated shot gradual transitions, vertically ternary structured video, and great intra-shot change in virtual scenes of game video.

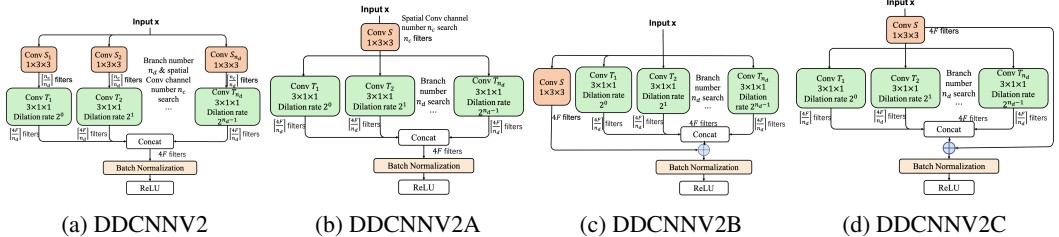

(a) DDCNNV2  (b) DDCNNV2A  (c) DDCNNV2B  (d) DDCNNV2C

Figure 3: Illustration of the search blocks for the first six blocks in the AutoShot. It consists of four types of factorized 3D convolution blocks, (a) DDCNNV2 with 2D spatial convolutions followed by 1D temporal convolutions, (b) DDCNNV2A with a shared 2D spatial convolution and 1D temporal convolutions, (c) DDCNNV2B accumulating a 2D spatial convolution and 1D temporal convolutions, and (d) DDCNNV2C compromising between DDCNNV2A and DDCNNV2B.

advanced 3D ConvNets (Qiu et al., 2017) and Transformers (Vaswani et al., 2017), we deign a shot boundary detection neural architecture search space. Specifically, AutoShot conducts architecture search on seven blocks, where we add a self-attention layer number search after the sixth block. The search blocks in the first six blocks are illustrated in Fig. 3. We firstly design four types of factorized 3D convolutions in the search space. Let $\mathbf{x}$ be the input for the current block, the four kinds of search blocks can be formulated as following:

(1) DDCNNV2: We conduct the search on the number of dilated convolution branches $n_d$, *i.e.*, 4 and 5, and the channel number of 2D spatial convolution $n_c$, *i.e.*, 1, 2 and 3 times of the input channel numbers, for the original DDCNNV2 block in the TransNetV2 as illustrated in Fig. 3a. The DDCNNV2 can be formulated as

$$\mathbf{h} = \text{ReLU}(\text{BN}(\text{Concat}([\mathbf{h}_1, \mathbf{h}_2, \cdots, \mathbf{h}_{n_d}]))), \; \mathbf{h}_i = (\mathbf{T}_i \cdot \mathbf{S}_i) \cdot \mathbf{x} = \mathbf{T}_i(\mathbf{S}_i(\mathbf{x})), \; i = 1, \cdots, n_d, \quad (1)$$

where $\mathbf{h}$ is the output of the current block, $\mathbf{S}_i$ is the 2D spatial convolution with the channel number $\lceil \frac{n_c}{n_d} \rceil$, $\mathbf{T}_i$ is the 1D temporal convolution with the channel number $\lceil \frac{4F}{n_d} \rceil$ and dilation rate $2^{i-1}$, and $F$ is a pre-defined channel number in Fig. 3a. The key components in DDCNNV2 are dilated temporal 1D convolutions, and factorized 3D convolutions with spatial 2D convolutions and temporal 1D convolutions. The design enforces diverse contextual temporal feature extraction and reduces the number of learnable parameters, which might reduce the over-fitting.

(2) DDCNNV2A: To unify the feature extractor of the spatial 2D convolutions, we can employ a shared 2D convolution instead of multiple branches of spatial 2D convolutions, as illustrated in

Fig. 3b. The shared spatial 2D convolution aims to extract a unified spatial feature for the following diverse temporal feature extractions. The DDCNNV2A can be expressed as

$$\mathbf{h} = \text{ReLU}(\text{BN}(\text{Concat}([\mathbf{h}_1, \mathbf{h}_2, \cdots, \mathbf{h}_{n_d}]))), \ \mathbf{h}_i = (\mathbf{T}_i \cdot \mathbf{S}) \cdot \mathbf{x} = \mathbf{T}_i(\mathbf{S}(\mathbf{x})), \ i = 1, \cdots, n_d, \quad (2)$$

where $\mathbf{S}$ is a shared 2D spatial convolution with searched channel number $n_c$.

(3) DDCNNV2B: Inspired by the design of Pseudo-3D network (Qiu et al., 2017), we construct another two search blocks to learn various spatio-temporal representations as illustrated in Fig. 3c and 3d. The DDCNNV2B can be given by

$$\mathbf{h} = \text{ReLU}(\text{BN}((\mathbf{S} + \mathbf{T}) \cdot \mathbf{x})) = \text{ReLU}(\text{BN}(\mathbf{S}(\mathbf{x}) + \mathbf{T}(\mathbf{x}))),$$
$$\mathbf{T}(\mathbf{x}) = \text{Concat}([\mathbf{T}_1(\mathbf{x}), \mathbf{T}_2(\mathbf{x}), \cdots, \mathbf{T}_{n_d}(\mathbf{x})]). \quad (3)$$

To ensure the channel numbers of spatial features and temporal features are equal, the channel number of 2D spatial convolution is fixed to be four times of the current block's input dimension number $4F$.

(4) DDCNNV2C: Different from DDCNNV2B, the temporal convolution of DDCNNV2C can still utilize the feature of spatial convolution as illustrated in Fig. 3d, which can be formulated as

$$\mathbf{h} = \text{ReLU}(\text{BN}((\mathbf{S} + \mathbf{T} \cdot \mathbf{S}) \cdot \mathbf{x})) = \text{ReLU}(\text{BN}(\mathbf{S}(\mathbf{x}) + \mathbf{T}(\mathbf{S}(\mathbf{x})))),$$
$$\mathbf{T}(\mathbf{S}(\mathbf{x})) = \text{Concat}([\mathbf{T}_1(\mathbf{S}(\mathbf{x})), \mathbf{T}_2(\mathbf{S}(\mathbf{x})), \cdots, \mathbf{T}_{n_d}(\mathbf{S}(\mathbf{x}))]). \quad (4)$$

We further construct a 1D temporal Transformer block after six factorized convolution layers to enhance the temporal modeling, whose input is a flattened frame-wise convolutional feature. We conduct the number of self-attention layers search $\{0, 1, 2, 3, 4\}$ in the Transformer block.

In summary, AutoShot has seven search blocks. In the first six search blocks, it conducts the channel number search and branch/dilation number search for DDCNNV2 and DDCNNV2A. Limited by the dimension number consistency of element-wise addition, DDCNNV2B and DDCNNV2C searches the branch number. Consequently, we have $3 \times 2 \times 2 + 2 \times 2 = 16$ options for each search block in the first six search blocks. The search space of AutoShot totally has $(16^6) \times 5 = 8.39 \times 10^7$ candidate architectures.

## 4.2 AutoShot Training and Search

After the construction of the base network for representational learning, we concatenate representations from the base network with RGB histogram similarity of raw input frame and learnable cosine similarity of concatenated block features (Chasanis et al., 2009; Souček & Lokoč, 2020) to construct a 4,864 dimension feature vector. Then a fully connected layer of 1,024 neurons with ReLU activation (Nair & Hinton, 2010) is added. Next a dropout layer (Krizhevsky et al., 2012) with a dropout rate 0.5 is used before the final two frame-wise classification heads for a single middle frame of a transition $y$ and all transition $z$.

The SuperNet training of AutoShot utilizes an efficient weight sharing strategy. The weight sharing strategy (Guo et al., 2020; Liu et al., 2018) encodes the search space in a SuperNet, and all the candidate architectures share the weights of the SuperNet. One shot NAS (Guo et al., 2020) decouples the SuperNet training and architecture search, which yields a better accuracy. We also conduct two sequential steps for SuperNet training and architecture search. We utilize a single path and uniform sampling strategy to reduce the co-adaptation between node weights (Guo et al., 2020).

We implement a Bayesian optimization (Shahriari et al., 2015) based architecture search for AutoShot. Bayesian optimization iterates between fitting probabilistic surrogate models and determining which configuration to evaluate next by maximizing an acquisition function. Gaussian process with a Hamming kernel is utilized as the surrogate function. We employ a random exploration in the initialization to obtain a good Gaussian process model. For the acquisition function, we use probability of feasibility (Gardner et al., 2014).

For the candidate architecture retraining, we firstly employ the same two classification, *i.e.*, single middle frame $y$ of a transition and all transition $z$, cross-entropy losses as the SuperNet training

$$\mathcal{L}_{retrain} = -\sum_{i=1}^{N} \sum_{j=1}^{N_F} [\lambda_1 y_{i,j} \log \hat{y}_{i,j} + \lambda_2 z_{i,j} \log \hat{z}_{i,j}], \quad (5)$$

where $N$ is the batch size in the stochastic gradient descent (SGD), $N_F$ is the pre-defined number of frames for each training sample which is processed in each batch of training, $\lambda_1$ and $\lambda_2$ are trade-offs between two classification heads, and $\hat{y}_{i,j}$ and $\hat{z}_{i,j}$ are two frame-wise predictions of single middle frame of a transition and all transition, respectively.

After retraining with the plain multi-head cross-entropy classification loss in Eq. equation 5, we further enhance the candidate networks by employing the best performing candidate network as a teacher network in knowledge distillation (Hinton et al., 2014), and utilize weight grafting (Meng et al., 2020) to further improve the best accuracy. The knowledge distillation is used to align candidate networks with a desired accuracy, and the weight grafting adaptively balances the grafted information among aligned networks, which improves the representation capability and boosts the accuracy. The entropy-based weight grafting can be formulated as

$$
\mathcal{L}(W) = -\sum_{i=1}^{N} \sum_{j=1}^{N_F} \left[ \lambda_1 \tilde{y}_{i,j} \log \hat{y}_{i,j} + \lambda_2 \tilde{z}_{i,j} \log \hat{z}_{i,j} \right],
$$

$$
\alpha = A \times (\arctan(c \times (H(W_l^{M_2}) - H(W_l^{M_1})))) + 0.5, \ W_l^{M_2} = \alpha W_l^{M_2} + (1 - \alpha) W_l^{M_1},
$$

(6)

where $\tilde{y}_{i,j}$ and $\tilde{z}_{i,j}$ are two frame-wise predictions of single middle frame of a transition and all transition from the teacher network, $A$ and $c$ are fixed hyperparameters, $\alpha$ is the coefficient based on entropy to balance networks, $H(\cdot)$ is the entropy based on the bins of network weights $W_l^{M_1}$ or $W_l^{M_2}$, $l$ is the layer in the network, $M_1$ and $M_2$ are two networks where network $M_2$ accepts the information from network $M_1$.

## 5 EXPERIMENTS

We conduct neural architecture search based on the evaluation metrics on the collected SHOT and validate the effectiveness of the searched optimal architecture on SHOT, ClipShots (Tang et al., 2018), BBC (Baraldi et al., 2015a), and RAI (Baraldi et al., 2015b). In both SuperNet training and candidate network retraining, we construct each training sample by concatenating two shots randomly and the number of frames $N_F$ in each training sample is set as 60. The hyperparameters $\lambda_1$, $\lambda_2$, $A$, $c$, the number of bins in the entropy calculation and the number of grafting networks in Eq. equation 6 are set as 5, 0.1, 0.4, 1.0, 10 and 3, respectively, which are generally followed the setting of previous works (Souček & Lokoč, 2020; Meng et al., 2020). We use stochastic gradient descent with learning rate 0.1, momentum 0.9, batch size 16 and the number of epochs 12. In the search, the number of populations per epoch is 48 and the total number of epochs is 100 with 20 epochs for the initialization. We utilize the OpenBox library (Li et al., 2021) to implement the Bayesian optimization in the search. We use one 32 GB NVIDIA Tesla V100 GPU for both SuperNet and candidate network training, and use eight 32 GB NVIDIA Tesla V100 GPUs to accelerate the search speed. All the code, data and trained models will be released for the reproducibility.

**Neural architecture search on SHOT** We train the SuperNet on the combined training set of SHOT and ClipShots, and conduct the search based on the evaluation metrics on the collected SHOT dataset. For a fair comparison, we closely follow the dataset protocol and metric calculation of previous work (Souček & Lokoč, 2020; Baraldi et al., 2015b). Additionally, we also conduct a search based on the precision metric given a fixed recall rate 0.71 same as TransNetV2, because a higher F1 metric cannot guarantee higher precision and recall scores simultaneously in practice. From Table 2 Left, AutoShot outperforms TransNetV2 by 4.2% and 3.5% based on F1 and precision metrics. Note that the F1 score of PySceneDetect (Castellano, 2022) on SHOT is less than 0.6, which is far behind AutoShot, and it can hardly handle challenging gradual transitions. For AutoShot, we find that 54% of missed shots are gradual transitions, and gradual transitions take 30% of shots in SHOT. Gradual transition has no huge inter-frame difference, which is difficult to be fully detected. In the following, we only compare AutoShot based on the F1 metric with other methods for consistency.

To compare the predictions visually, we employ video thumbnail images to clearly demonstrate the difference. The ground truth boundary is shown in the pink or light/white color as shown in Fig. 4. The detected boundary of AutoShot is marked as pink, cyan or light, and the detected boundary of TransNetV2 is visualized as cyan or light. AutoShot successfully detects minor shot transitions as shown in the pink color, where the TransNetV2 fails. For the clear transitions, both the TransNetV2 and AutoShot succeed, as shown in the light color. The false positives of both TransNetV2 and

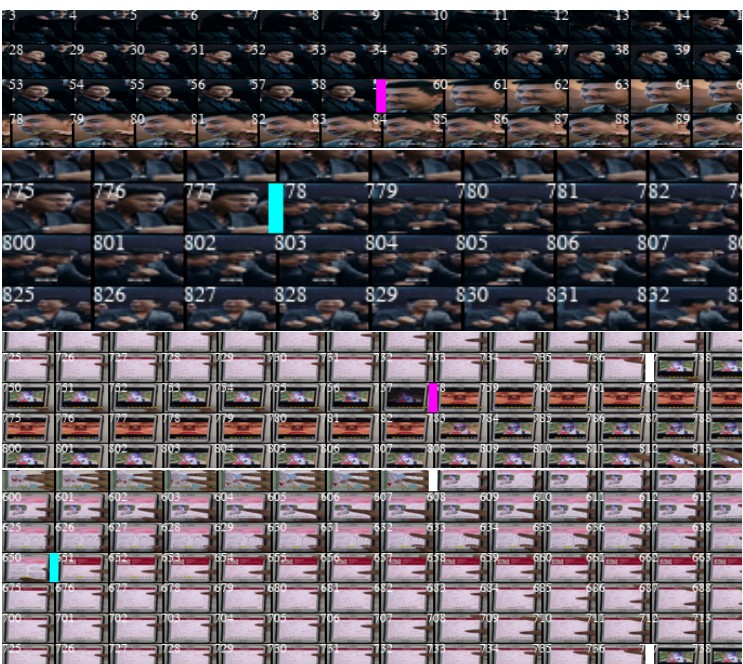

Figure 4: Visual comparison of shot boundaries of ground truth (pink, light/white), TransNetV2 (cyan, light/white), and AutoShot (pink, cyan, light/white) on four clips from SHOT dataset. AutoShot detects minor transition, shown in the pink color. The light/white color denotes that both the TransNetV2 and AutoShot successfully detects the shot boundary. The cyan denotes the false positives of both TransNetV2 and AutoShot, which might require adaptively understanding of contextual semantics across the video.

AutoShot are shown in cyan, which are hard negative shots. Reducing the false positives might require adaptively understanding of contextual semantics across the whole video, and some false positives are ambiguous even for human annotators.

**Generalization on other datasets** After obtaining the optimal network architecture from the SHOT dataset, we validate the network generalizability on other three publicly and widely used datasets, ClipShots (Tang et al., 2018), BBC (Baraldi et al., 2015a), and RAI (Baraldi et al., 2015b). The three existing datasets are quite different from our SHOT dataset, as shown in the section 3. BBC Planet Earth Documentary series (Baraldi et al., 2015a) consists of 11 episodes from the BBC educational TV series Planet Earth. Each episode is approximately 50 minutes long, and the whole dataset contains around 4900 shots and 670 scenes. RAI (Baraldi et al., 2015b) dataset is based on a collection of ten randomly selected broadcasting videos from the Rai Scuola video archive, where the length of each video is around half an hour. The ClipShots collects thousands of online conventional videos, not short videos, from YouTube, which is a much more challenging dataset than BBC and RAI. We use the same dataset split and protocol as previous work (Souček & Lokoč, 2020; Baraldi et al., 2015b).

From Table 2 Right, simply applying the searched AutoShot architecture to the three datasets obtains better F1 scores than previous state-of-the-art approaches consistently, which sufficiently validates the effectiveness and good generalizability of AutoShot. Specially, AutoShot outperforms previous state-of-the-art approaches by 1.1%, 0.9% and 1.2% on ClipShots, BBC and RAI. Note that we reproduce TransNetV2 based on PyTorch and obtain a F1 score of 0.776 on ClipShots, while the original paper (Souček & Lokoč, 2020) reports 0.779 based on TensorFlow.

**Effect of search space** We investigate the effect of three different search spaces, *i.e.*, AutoShot-S, -M and -L, in Table 3 (Left). AutoShot-S only employs DDCNNV2A components in the search space, which has six search options per block. AutoShot-M employs DDCNNV2 and DDCNNV2A components in the search space, which has 12 search options per block. The AutoShot-L denotes the search space defined in section 4.1, which achieves the best F1 score. The combined various 3D ConvNet variants, *i.e.*, DDCNNV2, DDCNNV2A, B and C, in each search block improves F1

| Method | TransNetV2 | AutoShot @F1 | AutoShot @Precision |
|--------|-----------|--------------|---------------------|
| F1 | 0.799 | **0.841** | 0.826 |
| Prec. | 0.904 | 0.923 | **0.939** |

| Dataset | ClipShots | BBC | RAI |
|---------|-----------|-----|-----|
| DSMs | 0.761 | 0.893 | 0.928 |
| ST ConvNets | 0.759 | 0.926 | 0.939 |
| TransNet | 0.735 | 0.929 | 0.943 |
| TransNetV2 | 0.776 | 0.962 | 0.939 |
| AutoShot | **0.787** | **0.971** | **0.955** |

Table 2: Left: AutoShot surpasses TransNetV2 by 4.2% and 3.5% based on the F1 score and precision metric with a fixed recall as TransNetV2. Right: the searched optimal architecture is validated on three widely used shot boundary detection datasets. AutoShot consistently achieves the best F1 compared to DSMs (Tang et al., 2018), ST ConvNets (Hassanien et al., 2017), TransNet (Lokoč et al., 2019) and TransNetV2 (Souček & Lokoč, 2020).

| AutoShot- | S | M | L |
|-----------|------|------|------|
| w/o retrain | 0.816 | 0.822 | 0.831 |
| w/ retrain | 0.833 | 0.837 | **0.841** |

| Method | w/o KD | w/ KD | w/ KD+ weight graft |
|--------|--------|-------|---------------------|
| F1 | 0.825-0.837 | 0.832-0.838 | **0.841** |

Table 3: F1 scores of different search spaces (Left), knowledge distillation (KD) and weight grafting (Right) on the collected SHOT dataset.

score in both retraining and candidate architectures after search, since the optimal blocks for different layers vary and more search options in AutoShot permit to identify the optimal composition.

**Searched architectures** Specifically, the obtained architecture based on F1 score, AutoShot@F1, is DDCNNV2$\{(n_c=4F, n_d=4)$, A$(n_c=4F, n_d=5)$, A$(n_c=4F, n_d=5)$, A$(n_c=4F, n_d=5)$, $(n_c=12F, n_d=5)$, $(n_c=8F, n_d=5)\}$, which has floating-point operations (FLOPs) of 37 GMACs. The obtained architecture based on precision metric, AutoShot@Prec., is DDCNNV2$\{(n_c=12F, n_d=4)$, $(n_c=8F, n_d=4)$, B$(n_d=4)$, C$(n_d=4)$, B$(n_d=5)$, B$(n_d=4)\}$, which has FLOPs of 30 GMACs. We find that the optimal architectures indeed employ diverse blocks and less number of operations than TransNetV2 of 41 GMACs. The search on SHOT chooses more dilated convolutional branches $n_d$ and diverse blocks, because more dilated convolutional branches $n_d$ and diverse blocks enhances the representational learning power for various temporal granular shot transitions, which vastly exist in short videos. This is probably another reason that AutoShot on SHOT achieves much bigger improvement than that on the conventional video datasets. Although the two optimal architectures use no self-attention layer, the training of SuperNet and architectures with close F1 or precision scores utilize the self-attention.

**Effect of knowledge distillation and weight grafting** We ablate knowledge distillation and weight grafting in Table 3 (Right) based on the constructed search space in section 4.1. Without knowledge distillation in Eq. equation 5, the range of F1 scores after retraining can be 0.825-0.837. We use the best performing, *i.e.*, F1 score of 0.837, architecture as the teacher network, and knowledge distillation aligns the candidate architectures with F1 score of range 0.832-0.838. Then, the weight grafting in Eq. equation 6 further improves the best F1 score by 0.3%.

## 6 CONCLUSION

In this work, we collect a new short video shot boundary detection dataset, named SHOT, which is a quite different scenario from conventional long video based shot boundary detection. The SHOT can significantly accelerate the development and evaluation of various short video based applications, *e.g.*, intelligent creation, and video scene segmentation and understanding. Leveraging this new asset, we propose to optimize the model design for the task of video shot boundary detection, by conducting neural architecture search in a search space encapsulating various advanced 3D ConvNets and Transformers. AutoShot surpasses previous best shot boundary detection method by 4.2% and 3.5% based on the F1 and precision scores, respectively. We further validate the generalizability of the searched optimal architecture on ClipShots, BBC and RAI. Experimental results demonstrate that, AutoShot has a good generalizability and outperforms previous state-of-the-art approaches on the three existing public datasets.

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
