# OpenReview forum: "AutoShot: A Short Video Dataset and State-of-the-Art Shot Boundary Detection"
_ICLR.cc/2023/Conference — Submitted to ICLR 2023_

### Official Review · Reviewer_3zbz · 2022-10-19

**Confidence:** 4
**Correctness:** 3
**Technical Novelty And Significance:** 2
**Empirical Novelty And Significance:** 2
**Recommendation:** 3

**Clarity, Quality, Novelty And Reproducibility:**

Please refer to the above for clarity, quality, and novelty. For reproducibility, I think a graduate student can reproduce it.


**Strength And Weaknesses:**

Strength:
- The paper shows its approach in detail, and the structure is formal.
- The experiments are tested on different datasets.

Weaknesses: But I have the following significant concerns on quality, and novelty:

- Dataset construction needs several steps, including data collection, annotation, and verification. The authors may need to clarify that the collection process is not challenging. The data may come from web crawling or database accessing if they are from a video company. And at the same time, the copyright is not protected. How to handle and protect the copyright of video creators is questionable.

- The annotation is challenging since it requires frame-wise, but it is not reasonable. The annotation approach that uses a thumbnail cannot provide enough information for the annotators to well handle the challenging cases. Instead, the results may be similar to conventional methods that detect shots based on appearance. The details of the annotation are not clear in the supplementary. How to ensure quality? With double check or voting?

- The method lacks novelty that does not show much difference from the exiting NAS papers on classification. The challenges mentioned by authors are actually very good points for design, e.g., incorporating an adaptive sliding window to handle the gradual transition, or separately handling the background and subject for intra-shot changes, or using rules of thumb for ternary videos. A well-designed lightweight mixture-of-expert network may well handle the challenges.

- The experiments are less interesting that cannot provide insights on the challenges mentioned by the authors, noting that the challenges are the motivation of the authors to bring out a new setting. And the qualitative results are misleading, and it is hard to tell the difference of different methods. And this also adds to my concerns about the quality of annotation.

Minor:
- What is the dataset scale comparison with the existing datasets?

**Summary Of The Paper:**

This paper studies shot boundary detection, a long-standing problem, in a particular case: short-form videos. The challenges come from complicated shot gradual transitions, vertically ternary structured videos, and intra-shot change in virtual scenes. The authors collect a new public Short video sHot bOundary deTection dataset, named SHOT, consisting of 853 complete short videos and 11,606 shot annotations. Based on this dataset, the paper proposes to optimize the model design for video SBD, by conducting a neural architecture search in a search space encapsulating various advanced 3D ConvNets and Transformers. It achieves higher F1 scores than previous state-of-the-art approaches.

**Summary Of The Review:**

Considering the novelty and quality of this paper, I think it does not reach the bar of the top-tier ICLR conference and vote for rejection.

---

> ### Author Response · Authors · 2022-11-17
> **Response to Reviewer 3zbz**
>
> Dataset collection: We agree that each step is not challenging. However, as agreed by the reviewer, the dataset collection consists of several steps, and we demonstrate these steps in the paper and supplementary. Specifically, the annotation process requires several rounds, back and forth between quality assurance / product team and annotation team. The data also needs to be carefully sampled to avoid bias on a specific category. Considering all these efforts and steps together, we believe building a good dataset, which requires the success of each step, is not an easy task. We spend over five months for this dataset, and split the dataset into three batches for the annotation. The annotation team labels the dataset for over four rounds just to make sure the annotation quality is good enough under the inspection of quality assurance team.
>
> Copyright: We have consulted an attorney, and received confirmation that the copyright is not an issue for the dataset. We can safely release the dataset and the dataset can be safely used by the public.
>
> Annotation: Frame-wise annotation means the annotation team views 960,794 images for one round annotation. In an image classification annotation task, the effort can build a large-scale image classification dataset.
>
> We agree, the thumbnail might be not enough to well handle the challenging cases. For the challenging cases, there are two ways to ensure the quality. One is the annotation is always inspected by the product team, and the annotation process iterates between the product team and annotation team. The other is, the annotation team can backtrack to the same timestamp in the original full resolution video when the annotator views a challenging case / transition. Thus, our results improve conventional appearance-based methods significantly, because we also consider the advanced temporal dependency. In the results, our AutoShot achieves 0.799 F1 score and conventional appearance-based method, PySceneDetect, achieves less than 0.6 F1 score.
> We do our best to ensure the high quality of the dataset through several rounds of iterations between product team and annotation team. The product team conducts the quality assurance through double-check the annotation, and will return the annotation task back to annotation team if over 5% inconsistency exists.
>
> Model design: In this work, we focus on an automated neural network design, which is a neural architecture search-based approach, for shot boundary detection. Manual design novel components and architectures is out of the paper’s scope. Thanks for the suggestion. We will manually design a novel architecture from the suggested perspectives in the future.
>
> Insight and qualitative results:
> There are two more insight findings from the results.
> 1 We find the AutoShot@F1 employs more dilation branches, n_d = 5, compared with AutoShot@Prec., n_d = 4. AutoShot@F1 with a higher dilation rate achieves a higher recall rate, while AutoShot@Prec. achieves a higher precision rate. A higher dilation rate leads to longer temporal modeling ability in AutoShot@F1, and reduces the false negatives. For these hardly detected shots, e.g., gradual transitions, increasing the modeling length helps the shot detection. AutoShot@Prec. instead focuses more on feature representation learning for an existing 3D cubic receptive field through more channel numbers for each dilation branch, especially much more channel numbers with n_c of 12F and 8F in the first two blocks. This reduces false positives. The neural architecture searched network configuration, which is totally and automatically learned from data, is reasonable and consistent with network design for false positives and false negatives reduction in shot boundary detection.
>
> 2 Compared with TransNetV2, AutoShot employs diverse components and has higher F1 and precision rates. Diverse blocks extract various aspects of representations. This enhances representation learning and is helpful for both false positives reduction and false negatives reduction.
>
> For the qualitative visualizations, please refer to response to Reviewer 6hJo for more details. Our AutoShot achieves better performance than previous methods because of the aforementioned advantages.
>
> Novelty: We believe our work is a good system, from dataset, robustness of method, to metric and evaluation. This paper collects the first dataset for short video boundary detection, conducts the first NAS based method for shot boundary detection, and achieves robust results and better accuracy than state-of-the-art approaches on four datasets. We believe that, the dataset can greatly accelerate the related method development, and the method sets a new baseline, which inspires the following studies.

---

### Official Review · Reviewer_wabR · 2022-10-24

**Confidence:** 4
**Correctness:** 4
**Technical Novelty And Significance:** 3
**Empirical Novelty And Significance:** 3
**Recommendation:** 6

**Clarity, Quality, Novelty And Reproducibility:**

The proposed method is novel and reproducible. However, some details of the method are not be clarified, as discussed in Weakness.

**Strength And Weaknesses:**

Strength:
1)	The newly released dataset SHOT is valuable, which will facilitate the development of community of short video understanding.
2)	The proposed neural architecture search based method is new for shot boundary detection, which outperforms previous SOTA approaches.

Weakness:
1)	It is difficult to understand for the readers without the neural architecture search background. It's technical contribution is limited.
2)	The hyper-parameter sensitivity analysis is missing in the experiments.
3)	The dataset is relatively small, which consists of 853 videos. A larger dataset will be more meaningful.


**Summary Of The Paper:**

For the task of shot boundary detection, this paper releases a new public dataset SHOT, which is complementary to existing related datasets. In addition, the model design is optimized by conducting neural architecture search in a search space encapsulating various advanced 3D ConvNets and Transformers. Experiments show the effectiveness of the proposed method.

**Summary Of The Review:**

This paper releases a new public dataset, as well as proposes an interesting method for shot boundary detection, which achieves SOTA. It is suggested to accept this paper.

---

> ### Author Response · Authors · 2022-11-17
> **We appreciate the valuable reviews!**
>
> We will try our best to add more comments and maintain the repository to help readers use and understand our work.
>
> In the paper, we evaluate the hyper-parameters of, with and without knowledge distillation, with and without weight grafting, three different search spaces, generalization ability of the AutoShot. For other hyper-parameters, such as lambda_1, lambda_2 in Equation (5), A and c in Equation (6) are generally following previous related work, because running one video experiment is expensive and costs a long time.

---

### Official Review · Reviewer_6hJo · 2022-10-27

**Confidence:** 4
**Correctness:** 3
**Technical Novelty And Significance:** 3
**Empirical Novelty And Significance:** 3
**Recommendation:** 6

**Clarity, Quality, Novelty And Reproducibility:**

The paper is well-written and easy to follow. In my view the work is original as there is no existing dataset of such. The authors promise to release the code and data so hopefully the results are reproducible.

**Details Of Ethics Concerns:**

The data are user generated videos, which may inevitably contain private information of personnels. I do recommend that the authors can describe how they handle the privacy issues in the revised version of this paper.

**Strength And Weaknesses:**

Strengths:
- The dataset fills the blank for shot boundary detection on short videos. Previous datasets do not contain the vertically viewed and short videos which is very popular these days.
- Apparently the AutoShot model achieves state-of-the-art performance on all three benchmarks, ClipShots, BBC and RAI.

Weaknesses:
- Size of the SHOT dataset seems small, as it contains 853 videos and most of which are shorter than 1 minute. A few reference dataset:
  - ClipShots, 4039 videos with 128636 cut transitions and 38120 gradual transitions. Length of most videos is greater than 2 minutes.
  - Short Video Dataset (SVD, Jiang et al. 2019), over 500,000 short videos and 30,000 labeled pairs of near-duplicate videos. SVD is designed for the video retrieval task.
- In terms of the proposed model, there is no design that is specifically motivated by the unique challenges that short video pose. Instead, NAS is performed on top of existing state-of-the-art work. In other words, the model design is not well motivated but more through brute-force engineering upon existing models.
- It's unclear if the challenges mentioned in Figure 1 have been well tackled by the AutoShot model. How well does the best performing model do on the challenging cases described in Figure 1? Are there some qualitative demos on each type of challenge?

**Summary Of The Paper:**

This paper introduces a new dataset, SHOT, for short video shot boundary detection (SBD). Together with the dataset, the authors propose a baseline model, AutoShot, through NAS among 3D ConvNets and Transformers components. The authors have demonstrated the unique statistics of the SHOT dataset, arguing that SHOT can be used to solve challenges of short video SBD. The AutoShot model is evaluated on three public benchmarks against previous state-of-the-art approaches, and the performance is promising.

**Summary Of The Review:**

Overall this is a good paper with a hopefully useful dataset that the video community can benefit from, especially due to the overwhelming trend of short and vertical videos. Although obtained by NAS on top of existing modules, the AutoShot model sets new standard of performance on multiple public benchmarks.

---

> ### Author Response · Authors · 2022-11-17
> **Thank you so much for the helpful reviews!**
>
> We are also keeping doing related research, such as scene boundary detection, and action detection and localization. We also collect more data and expect to release more data when we finish these projects. On the other hand, the shot boundary detection is evaluated in the shot level and 11,606 shots can be a medium dataset for model evaluation. Lastly, our collected SHOT dataset is the first dataset for short video shot boundary detection, and we expect more related work and datasets coming out in this direction in the nearby future.
>
> For the qualitative visualizations, we compare AutoShot with TransNetV2 in Figure 4 and supplementary. In Figure 4, the first transition, shown in pink color, is the quick transition, which is the second challenge shown in Figure 1. And our AutoShot can successfully detect it, while TransNetV2 misses the transition. The fourth boundary in Figure 4, shown in pink color, is a combination of several complicated gradual transitions, including color and brightness changes, which is the first challenge shown in Figure 1. Our AutoShot can still successfully detect it, while TransNetV2 misses it. There are more cases in the supplementary.
> Compared with TransNetV2, AutoShot employs diverse components and has higher F1 and precision rates. Diverse blocks extract various aspects of representations. This enhances representation learning and is helpful for both false positives reduction and false negatives reduction.
>
> We also compare our AutoShot with a widely used non-deep learning-based approach, PySceneDetect. The F1 score of PySceneDetect on SHOT is less than 60%, which is far behind AutoShot, and it can hardly handle gradual transitions.

---

> ### Comment · Program_Chairs · 2022-11-28
> **msg from SPC**
>
> Authors - please comment on ethics concern raised above.
>
> AC - if the issue isn't resolved, please bring it back to us.

---

> > ### Author Response · Authors · 2022-11-29
> > **Thank you so much for the reviews!**
> >
> > Dear SPC,
> >
> > To remove human private information, we employ a state of the art face detector [1, 2] to detect and obscure the human face region, as demonstrated in the figures of the paper.
> >
> > Several authors then also manually checked each individual video independently, in order to make sure the quality and integrity (no face privacy leakage, no NSFW content) of all these videos.
> >
> > Thank you so much for the reviews!
> >
> > [1] Detect Faces Efficiently: A Survey and Evaluations, IEEE Transactions on Biometrics, Behavior, and Identity Science, 2021
> >
> > [2] A Systematic IoU-Related Method: Beyond Simplified Regression for Better Localization, IEEE TIP 2021

---

### Official Review · Reviewer_Ar2e · 2022-11-01

**Confidence:** 3
**Correctness:** 4
**Technical Novelty And Significance:** 2
**Empirical Novelty And Significance:** 3
**Recommendation:** 5

**Clarity, Quality, Novelty And Reproducibility:**

Overall, the paper is easy to follow and clearly written (see above for some suggestions for improving the organisation of the paper).
The experimental part looks solid.
The main novelty of the paper is the new benchmark.
Given the fairly involved multi-step pipeline, I wouldn't be confident in being able to reproduce the results from the paper. But the authors state that they will release both the dataset and source code so reproducibility shouldn't be an issue.

**Strength And Weaknesses:**

Strengths:
- Provides a dataset/benchmark for shot boundary detection on short videos.
- Solid baseline with a model architecture that is specialised for shot boundary detection.

Questions:
- Boundary detection is crucial for long videos. Is this still true for short videos or would a less pipelined/more end-to-end approach be more appropriate, the more as "the frame-wise shot boundary annotation is a heavy task"?
- Is this task relevant for a broader/broad enough audience?

Organisation of paper:
- Section 3.2: It would be useful if this would be the place with all data-related statistics, including frame size, FPS, type/format of annotation, etc. In the current version, the information is spread across the abstract, introduction, appendix.
- Section 4: I suggest to move the details on the basic shot boundary detection model from Section 4.2 into a separate paragraph (e.g. 4.0).

**Summary Of The Paper:**

The paper is about shot boundary detection of short-form videos (cf. TikTok or YouTube Shorts).
According to the authors, short videos are one of the most prevailing medias in these days and shot boundary detection is a fundamental task for many following semantic understanding tasks.
More specifically, the paper carefully motivates the need for an additional dataset for this task, and announces the new dataset "SHOT" along with a baseline (AutoShot).
The baseline is the outcome of the neural architecture search over a search space spanned by existing architectures (3D ConvNets and Transformers), specialised for shot boundary detection.
The such searched model gives a gain of 4% on the new dataset and 1% on other public benchmarks.

The main contributions of the paper are the new benchmark with a highly competitive baseline (accuracy-wise and compute-wise) found by a neural architecture search.


**Summary Of The Review:**

On one hand, the recent popularity of short videos comes with the need for new public benchmarks to facilitate development and comparison of models, and the paper fills in on this.
On the other hand, the contribution in terms of modelling, algorithms or analysis is limited.
Which makes my recommendation "borderline".

---

> ### Author Response · Authors · 2022-11-17
> **Response to Reviewer Ar2e**
>
> 1 Boundary detection is crucial for both short videos and long videos. First, the average length of a short video is about 2–3 minutes, while the average length of a shot is about 2 seconds. Thus, one short video has averagely 60 shots. Actually, for the standard video Transformers, e.g., Video Swin and MViT, the input number of frames is 32 with sampling rate of 3, which equals to 3 seconds. And directly handling 2 – 3 minutes short video is still challenging. Shot can be a basic element, even for short video processing.
>
> Second, as mentioned in the paper, in order to obtain appealing, vivid and persuasive effects, the short video creators try to compress the content into a short length video, which dramatically leads to significant content, display, temporal dynamics and shot transition differences. Therefore, the short video-based shot boundary detection is much more challenging.
>
> Third, the shot boundary detection is a necessary and essential prequisite task for other applications such as scene boundary detection, action detection, intelligent creation, action classification and anomaly detection in a video. For instance, in the intelligent creation, if the desired video length is 30 seconds and the provided raw video has a length of 3 minutes, one fundamental step is to split the short video into several clips employing shot boundary detection.
>
> 2 This paper is relevant for a broader enough audience. First, as aforementioned, the shot boundary detection is crucial for both short and long videos. This paper can be a fundamental component for other video processing applications. Second, the dataset created in the paper can be used as an evaluation dataset for almost all spatial-temporal or structural modeling approaches, especially boundary detection and action detection. Third, we validate a mature solution for costly spatial-temporal architecture search which includes diverse 3D convolution blocks, self-attention blocks, knowledge distillation and grafting. The algorithm can be easily applied to other spatial-temporal modeling tasks.
>
> 3 Thank you so much for the suggestions! We will move all data-related statistics into Section 3.2 and add a separate paragraph to describe the basic shot boundary detection method.

---

> > ### Comment · Reviewer_Ar2e · 2022-11-22
> > **Response acknowledged**
> >
> > Thanks for the clarification.
> > A general observation (NLP is an excellent example, I think) is that breaking down the actual task into smaller and smaller subtasks doesn’t necessarily simplify the task: (i) sub-tasks may be harder to solve in isolation (in particular, the more challenging the conditions are, cf. your response 1/second paragraph); (ii) accumulation of errors you can’t recover from; (iii) annotation can be challenging (see also concern by other reviewer), etc.
> > I understand that video processing comes with its own challenges and the state of the art isn't the same for different modalities.
> > In this sense I agree that this work may be a stepping stone for further development of short video analysis techniques.

---

### Decision · Program_Chairs · 2023-01-20

**Decision:**

Reject

**Justification For Why Not Higher Score:**

- Very limited dataset size
- Limited technical contribution and novelty

**Justification For Why Not Lower Score:**

N/A

**Metareview: Summary, Strengths And Weaknesses:**

Authors propose a new dataset for short video shot boundary detection (SHOT) and a corresponding baseline model obtained with neural architecture search. Authors argue that short videos are among the most popular media sources which renders the problem highly significant. The dataset consists of 853 short videos and approximately 11K shot annotations. The neural architecture for the baseline model is performed over a search space consisting of various 3D CNNs and Transformer models. The authors show that in this setting the model outperforms prior work in terms of the F1 score, both on SHOT and several existing datasets.

The reviewers appreciated the timeliness of this research given the popularity of short videos -- there is a clear need for new public benchmarks to facilitate development and evaluation of these models. However, the reviewers also identified several points for improvement. (1) Limited dataset size, hence limited diversity. (2) Model design seems to be closer to a brute-force approach combining existing models, rather than a well-motivated method for handling short videos. (3) Limited technical contribution and novelty. (4) Organisation and clarity of presentation. While some of these can be addressed in a revised version, the rebuttal didn't adequately address the main challenges and I will hence recommend rejection.